# Ocean convection linked to the recent ice edge retreat along east Greenland

Kjetil Våge [1], Lukas Papritz [1], Lisbeth Håvik [1], Michael A. Spall[2] & G.W.K. Moore [3]

Warm subtropical-origin Atlantic water flows northward across the Greenland-Scotland Ridge into the Nordic Seas, where it relinquishes heat to the atmosphere and gradually transforms into dense Atlantic-origin water. Returning southward along east Greenland, this water mass is situated beneath a layer of cold, fresh surface water and sea ice. Here we show, using measurements from autonomous ocean gliders, that the Atlantic-origin water was re-ventilated while transiting the western Iceland Sea during winter. This re-ventilation is a recent phenomenon made possible by the retreat of the ice edge toward Greenland. The fresh surface layer that characterises this region in summer is diverted onto the Greenland shelf by enhanced onshore Ekman transport induced by stronger northerly winds in fall and winter. Severe heat loss from the ocean offshore of the ice edge subsequently triggers convection, which further transforms the Atlantic-origin water. This re-ventilation is a counterintuitive occurrence in a warming climate, and highlights the difficulties inherent in predicting the behaviour of the complex coupled climate system.

[1] Geophysical Institute and Bjerknes Centre for Climate Research, University of Bergen, Bergen 5007, Norway. [2] Woods Hole Oceanographic Institution, Woods Hole 02543 MA, USA. [3] University of Toronto, Toronto, ON M5S A17, Canada. Correspondence and requests for materials should be addressed to K.V. (email: kjetil.vage@uib.no)

Without redistribution of heat by the atmospheric circulation and ocean currents, only a small portion of the Earth's surface would be habitable. In the Atlantic Ocean the poleward transport of heat is largely accomplished by the Atlantic meridional overturning circulation (AMOC), in which warm waters are transported northward near the surface and cold waters are returned to the south at depth[1]. Most of the North Atlantic deep waters originate from the Nordic Seas[2], as result of a warm-to-cold transformation that takes place primarily in the eastern part of that region[3–5]. The resulting product, referred to as Atlantic-origin water[6], is returned southward by the East Greenland Current[7] (Fig. 1). This is a main source of dense water to the overflow plume that passes between Iceland and Greenland through Denmark Strait[8] and provides the largest and densest contribution to the lower limb of the AMOC[9].

The other major contribution to the Denmark Strait overflow is Arctic-origin water formed in the interior Iceland and Greenland Seas[10]. Proximity to the ice edge, where the highest ocean to atmosphere fluxes of heat occur due to frequent intense cold air outbreaks in which frigid, dry air masses are advected over comparatively warm surface water, is an important factor[11–14]. Diminished heat loss from this region owing to a retreat of the ice edge and different rates of warming of the ocean and the atmosphere will likely reduce the formation of Arctic-origin water[15]. In the Iceland Sea this water mass is primarily produced on the north-western outskirts of the cyclonic gyre, where the atmospheric forcing is more intense[13,16]. Farther to the west (within the region outlined in black in Fig. 1), the influence of fresh, low-density surface water becomes dominant[17,18]. The presence of this water mass is known to inhibit convection[19,20], but thus far a lack of wintertime measurements from the westernmost part of the Iceland Sea has prevented verification[13]. Here we show that this fresh, low-density surface layer is not present in the western Iceland Sea in winter, which permits the formation of deep mixed layers off-shore of the ice edge. As the ice edge retreats toward Greenland, water masses including the Atlantic-origin water transported by the East Greenland Current that were previously insulated from the atmosphere underneath sea ice are now being ventilated.

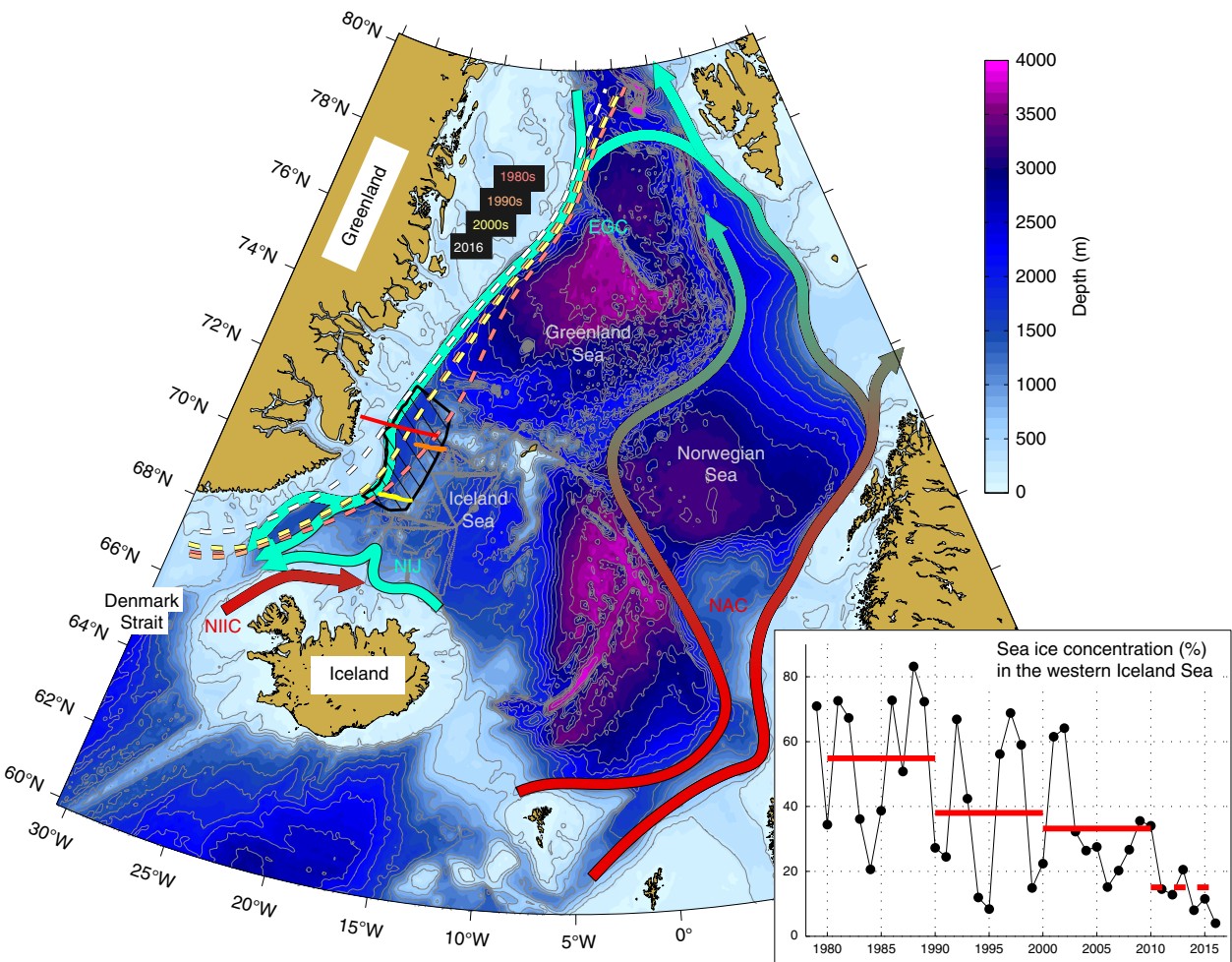

**Fig. 1** Schematic diagram of the currents that supply dense water to Denmark Strait. The colours indicate warm (red) to cold (green) transformation. Fresh, low-density water also flows along the Greenland continental slope (not shown). Hydrographic profiles from the western Iceland Sea region, outlined in black, document the seasonal cycle of the fresh surface layer (Fig. 4). The red line represents a hydrographic section obtained from a research ship in August 2012. The grey lines are glider trajectories and the orange and yellow lines indicate transects from the two gliders that were operating in the western Iceland Sea in February and April 2016, respectively. The dashed lines along the Greenland shelf represent the mean 50% sea ice concentration[38] contours from winter (Jan–Mar) 2016 (white) and decadal means from 1980–1989 (red), 1990–1999 (orange), and 2000–2009 (yellow). The acronyms are: EGC = East Greenland Current; NIJ = North Icelandic Jet; NIIC = North Icelandic Irminger Current; NAC = Norwegian Atlantic Current. The inset shows the winter mean sea ice concentration within the western Iceland Sea region outlined in black. The red lines are decadal means, and the dashed red line represents the mean from 2010 to 2016

## Results

**Re-ventilation of Atlantic-origin water along east Greenland.**
In the summer and fall of 2015 three autonomous gliders (see the
Methods section for details of the processing and calibration of
the glider data) were deployed with the purpose of mapping the
extent of convection in the Iceland Sea during the subsequent
winter. Here we focus on measurements obtained near the ice
edge in the western Iceland Sea. Vertical sections of potential
temperature and salinity from a cruise in summer 2012 and the
two glider transects that approached the ice edge in winter
2015–2016 are shown in Figs. 2 and 3. The summer section has a
pronounced 50–100 m thick fresh, low-density layer at the surface
and a well-defined Atlantic-origin water mass beneath char-
acterised by intermediate maxima in temperature and salinity
(Figs. 2a and 3a). By late winter the surface layer had vanished
and only a trace of the intermediate temperature maximum is
visible ($\theta \geq 0$ °C). Instead, the western end of the February 2016
glider transect has an approximately 400 m deep mixed layer with
a density of $\sigma_\theta = 28.01$–$28.02$ kg/m³, more than sufficient to
supply the dense Denmark Strait overflow plume (Figs. 2b and
3b). As this was still relatively early in the convective season, most
likely the depth and density of the mixed layer increased further
into winter due to continued heat loss to the atmosphere[13]. The
second glider encountered a re-stratifying water column farther to
the south in late April, at the end of the convective season, that
also shows similarly modified Atlantic-origin water (Figs. 2c and
3c). Both gliders were turned around due to the proximity of sea
ice prior to reaching the core of the East Greenland Current.
Hence, the extent to which the bulk of the Atlantic-origin water
mass was further transformed by convection and how readily the
modified waters were transported toward Denmark Strait remain
uncertain. However, the glider transects clearly demonstrate that

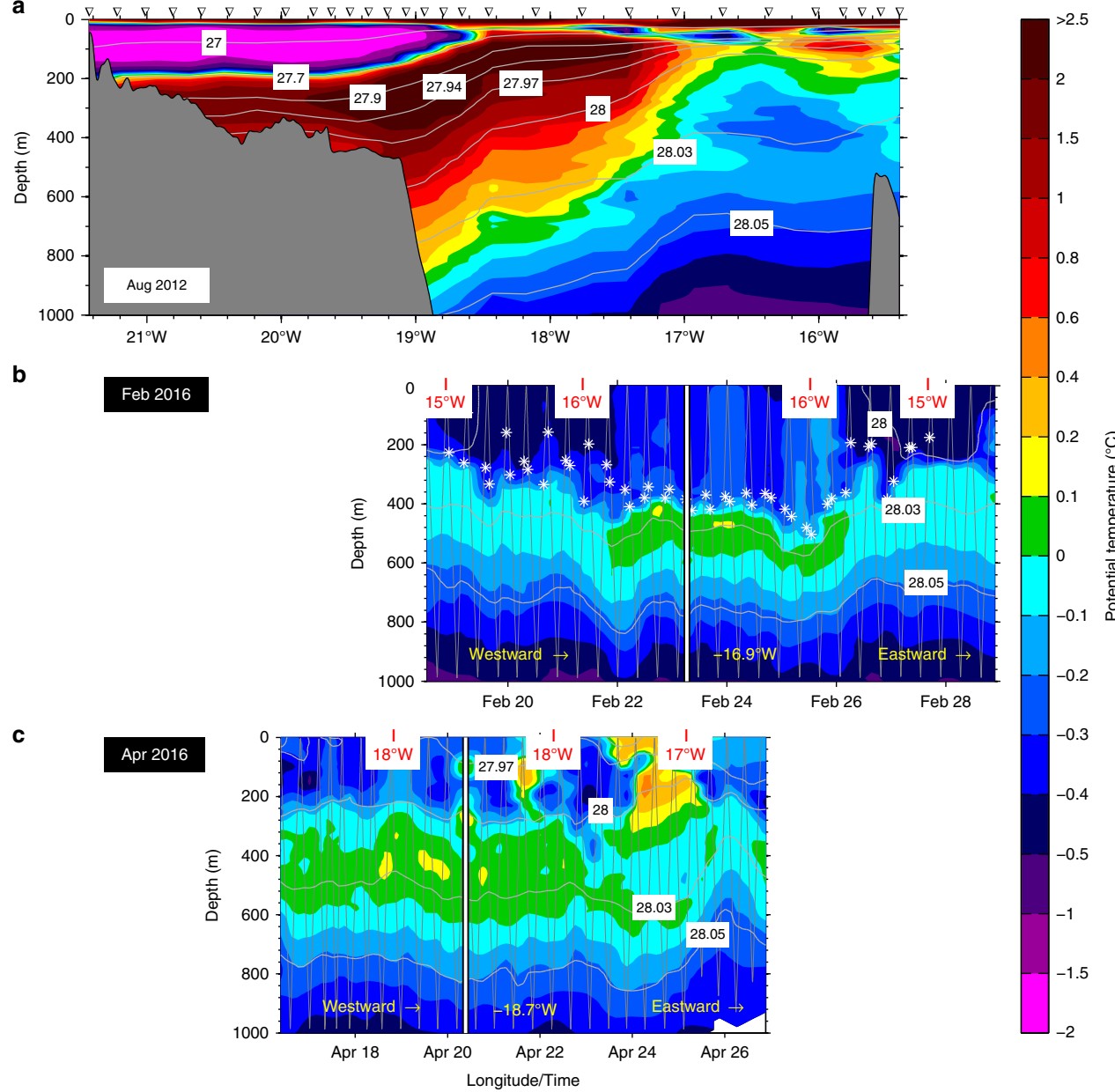

**Fig. 2** Potential temperature transects off east Greenland. Vertical sections (**a**–**c**) from the shipboard and glider measurements indicated in Fig. 1. The
gliders moved westward toward the ice edge, turned around at the longitude indicated by the white lines that are aligned with the shipboard transect, and
then returned eastward along the same trajectory. The white crosses mark the depth of the mixed layer in the February 2016 glider transect (**b**)

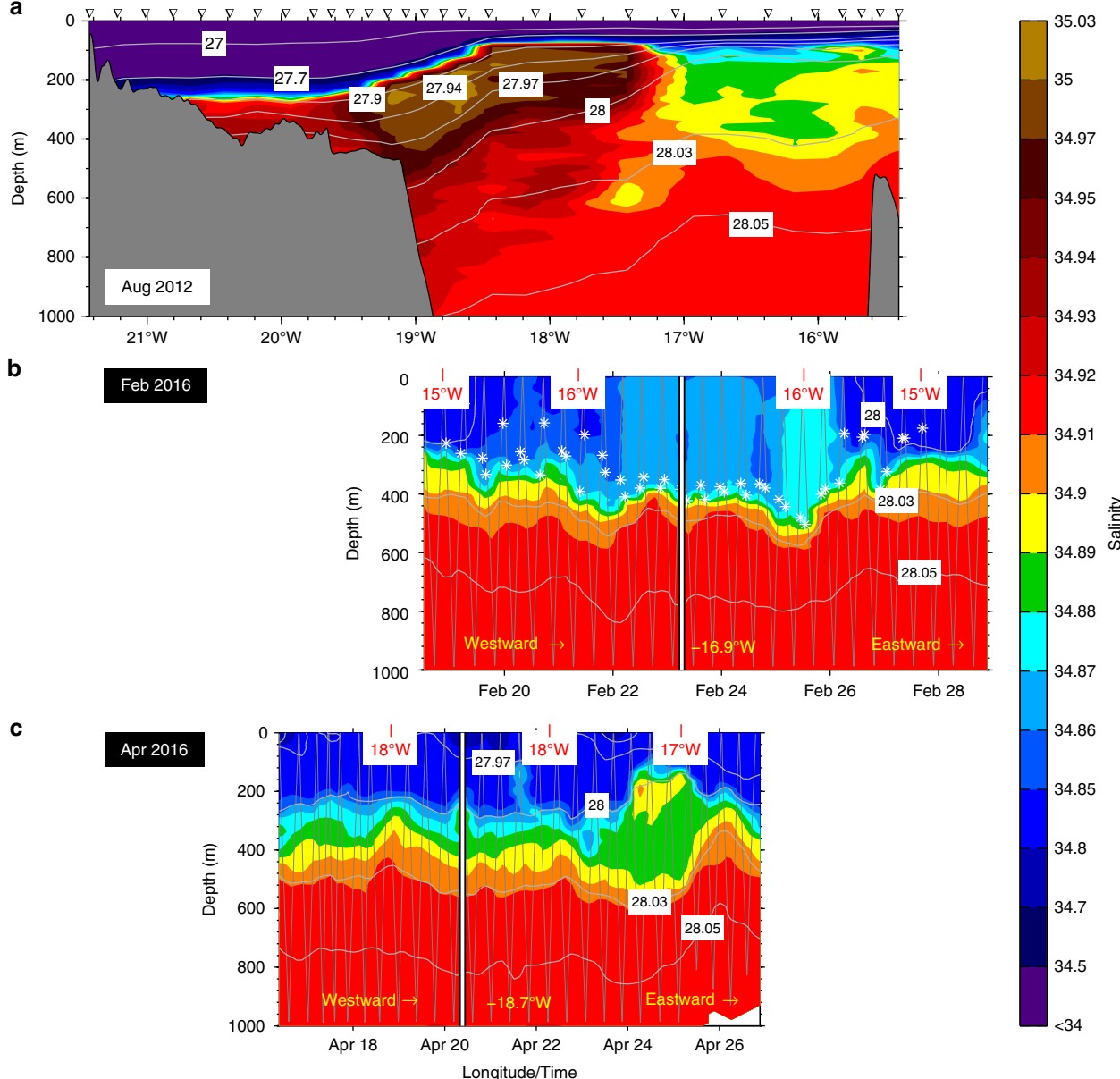

**Fig. 3** Salinity transects off east Greenland. Vertical sections (**a**–**c**) from the shipboard and glider measurements indicated in Fig. 1. The gliders moved westward toward the ice edge, turned around at the longitude indicated by the white lines that are aligned with the shipboard transect, and then returned eastward along the same trajectory. The white crosses mark the depth of the mixed layer in the February 2016 glider transect (**b**)

the fresh surface layer characteristic of the western Iceland Sea in summer does not prevent wintertime convection off the ice edge and indicate that the Atlantic-origin water transported by the East Greenland Current is re-ventilated by direct contact with the atmosphere while transiting this region in winter.

The western Iceland Sea is a sparsely sampled region, in particular in winter. Nonetheless, using newly available data from instrumented seals[21] in combination with historical measurements[13], the monthly mean hydrographic properties averaged over the upper 50 m of the water column demonstrated a pattern that is consistent with the shipboard and glider measurements (Fig. 4). The fresh surface layer starts developing in May or June and is most pronounced between July and August with a salinity well below 34. In October the surface layer becomes substantially more saline, and by November a well-defined surface layer is no longer present. At that time the water column has an ~80 m deep mixed layer and is well preconditioned for convection.

**Local formation of deep mixed layers**. In order to investigate whether the deep mixed layers observed at the western turn-around point of the February glider transect may have formed locally, a one-dimensional model (see the Methods section and ref. [22] for details of the model) was employed. The August 2012 profile closest to the turnaround point was used as initial conditions and a constant heat loss of 120 W/m², typical for winter 2015–2016 (Fig. 5b), was applied from November to mid-February. As expected from previous work in this region[19,20], the fresh surface layer prevented convection, resulting in effect instead in the formation of sea ice as soon as the cooling commenced. This demonstrates that the disappearance of the fresh water is not the result of vertical mixing caused by heat loss to the atmosphere. Another mechanism is instead required to remove the freshwater prior to the onset of convection.

Strong northerly winds are prevalent along the east coast of Greenland in fall and winter[23]. We estimate the onshore Ekman

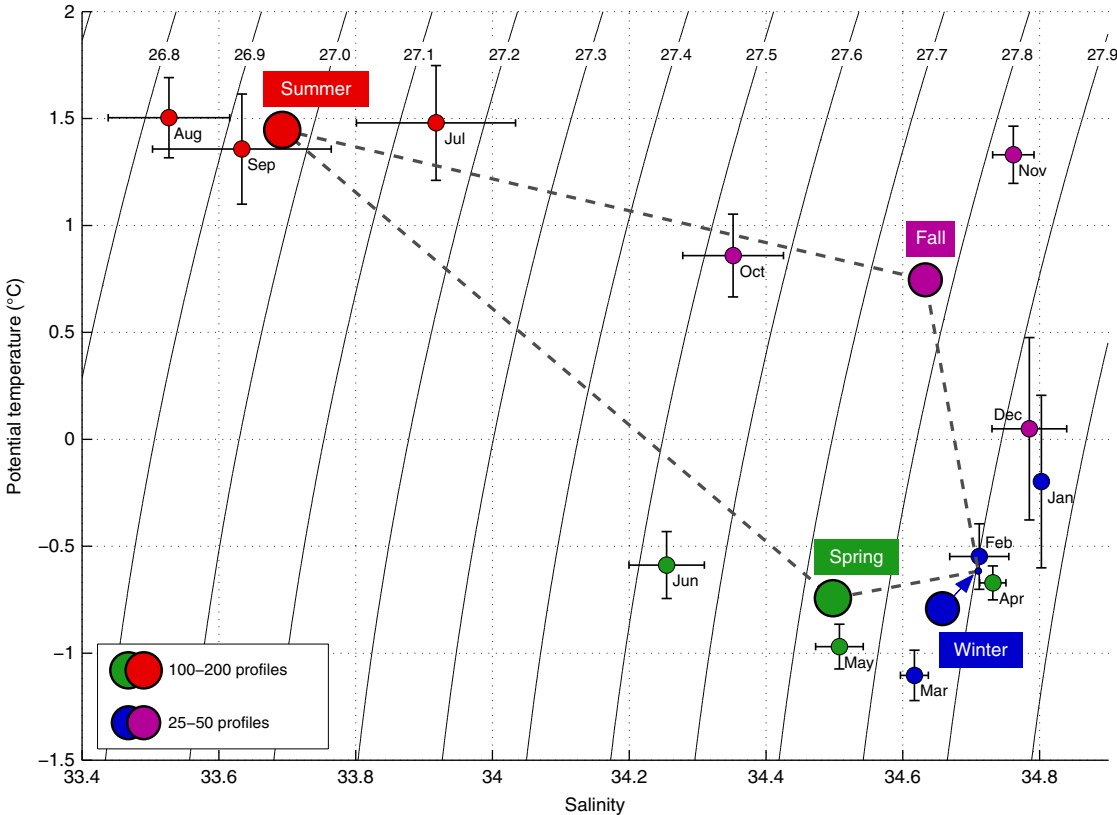

**Fig. 4** Upper-ocean hydrographic properties in the western Iceland Sea. Monthly and seasonally averaged temperature and salinity from the upper 50 m of the water column within the region outlined in Fig. 1. The black lines represent standard errors. The blue arrow indicates the true position of the winter mean, which has been offset in order to reduce clutter

transport induced by these northerly winds by assuming an Ekman layer depth of 50 m, and find a pronounced seasonal cycle (Fig. 5a). While there is no substantial Ekman transport in summer, it increases considerably through fall. We estimate that the Ekman transport is sufficient to flush the fresh, low-density layer onto the Greenland shelf, thereby preconditioning the western Iceland Sea region for convection before the atmospheric forcing peaks between December and March (Fig. 5b). The integrated Ekman transport distance through the strong forcing season of order 300 km sets the horizontal scale of the band that is preconditioned for convection. We note that such northerly barrier winds, which often occur along the entire east coast of Greenland in winter[24,25], inhibit the offshore diversion of freshwater and hence limit its impact on convection in the interior Nordic and Irminger Seas. In 2015–2016 the non-linear combination of unusually strong northerly winds and high heat fluxes at the beginning of winter, along with nearly ice-free conditions, were particularly conducive for convection (Fig. 5). Melting of sea ice in summer, when onshore Ekman transport is negligible, helps replenish this fresh surface layer[26].

A new simulation with the one-dimensional model was initialised in which the fresh surface layer was replaced by a mixed layer extending from 80 m depth to the surface, in agreement with the mean November profile from the combined seal and historical hydrographic data set. The resulting simulated profile (red trace in Fig. 6) corresponds very closely to the profile recorded by the glider at the turnaround point in February 2016 (blue trace in Fig. 6), even though the initial conditions were obtained from summertime measurements 4 years prior to the glider transect. This implies that the deep and dense mixed layer recorded by the glider in February 2016 is consistent with local formation as the result of convection offshore of the ice edge.

## Discussion

A consequence of the re-ventilation of Atlantic-origin water east of Greenland may have been detected farther downstream. At the Denmark Strait sill there is no discernible seasonal variability in the overflow transport, but a significant seasonal signal in temperature. In particular, the temperature of the Denmark Strait overflow is minimum in September[27], which, assuming an advective speed of 4–5 cm/s, is consistent with the transport of colder Atlantic-origin water that was re-ventilated the previous winter in the Iceland Sea. Taking into account that the re-ventilation may also have taken place farther north along the ice edge east of Greenland, this is in reasonable agreement with observed velocities of the Atlantic-origin water in the East Greenland Current[7,8].

The sea ice concentration in the western Iceland Sea has diminished substantially over the last few decades[15]. There was, in particular, very little sea ice in winter 2015–2016 relative to climatological values (Fig. 5c). As a consequence, increasing areas along the Greenland continental slope, and hence also of the East Greenland Current, are no longer insulated by sea ice from interaction with the atmosphere in winter. If the ice edge continues to retreat toward Greenland, we may expect to see more re-ventilation of Atlantic-origin water and higher heat fluxes into the atmosphere along the pathway of the East Greenland Current from Fram Strait to Denmark Strait. If such re-ventilation becomes more pronounced, this could further modify the properties of the Denmark Strait overflow and thereby impact the lower limb of the AMOC. While large-scale sea ice loss may lead to a weakening of the AMOC[28], the ice edge retreat toward Greenland causing increased ventilation of Atlantic water in the Nordic Seas during a warming climate is unexpected and contrary

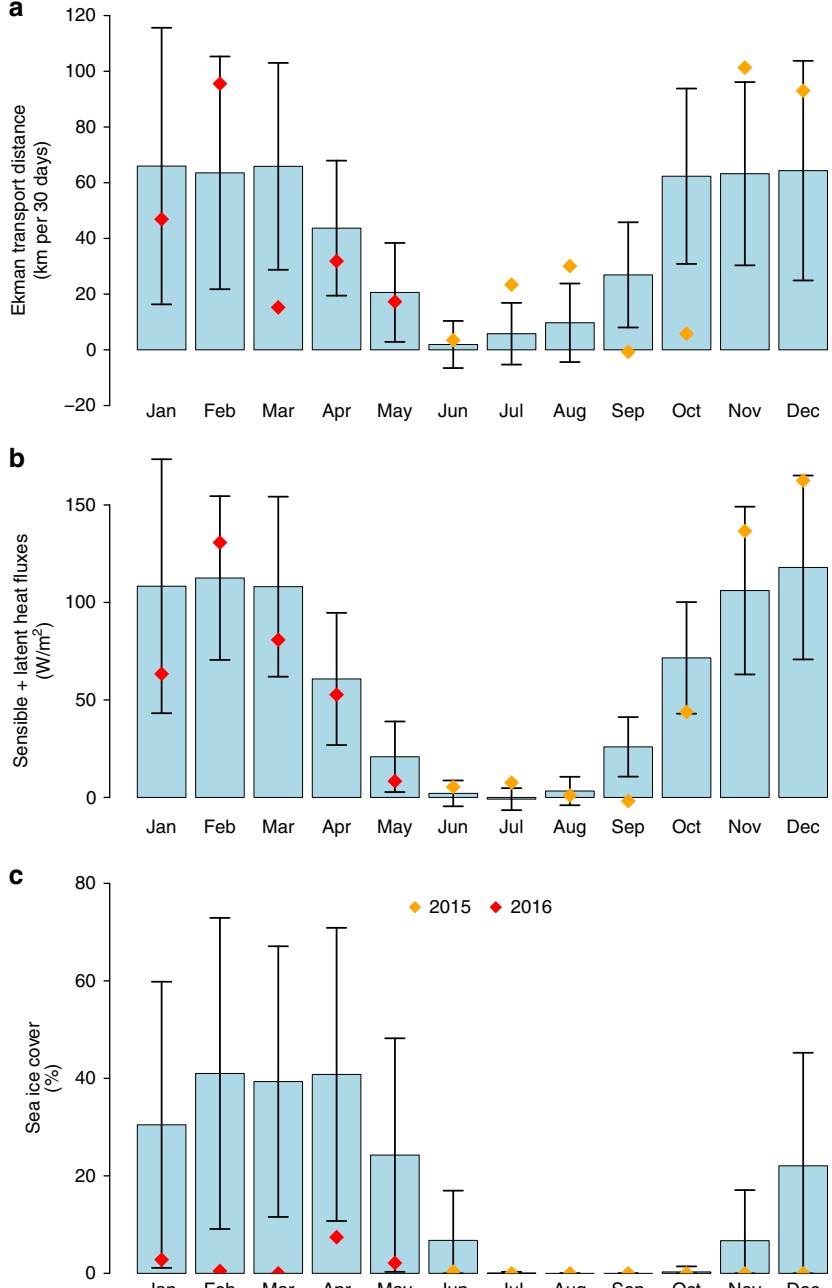

**Fig. 5** Atmospheric and sea ice conditions in the western Iceland Sea. The panels show **a** monthly mean Ekman transport distances (positive direction toward the west, Ekman layer depth taken to be 50 m), **b** monthly mean total turbulent (sensible + latent) heat fluxes, and **c** monthly mean sea ice concentrations in the region outlined in Fig. 1 over the ERA-Interim period (1979–2016). The black lines represent the standard deviations, and the red and yellow squares mark the corresponding values from the 2015–2016 deployment. Wind stress and turbulent heat fluxes are averaged over the ice-free portions of the region

to the tendency of reduced convection generally predicted by climate models[29].

## Methods

**Hydrographic data**. The wintertime transects were obtained by autonomous, buoyancy-driven Seagliders that are capable of diving to 1000 m depth[30]. The gliders profile in a sawtooth pattern with a typical vertical to horizontal glide ratio of 1:3 that results in a speed through water of about 20 cm/s. Conductivity and temperature were sampled at 20–80 s intervals, resulting in a vertical resolution of 1–5 m, enhanced within the surface mixed layer. Following a tuning of the flight model parameters for each glider, the data were reprocessed and corrections were applied to compensate for thermal-inertia and flushing speed issues arising from usage of unpumped sensors[31]. Temperature and salinity measurements outside the expected range of values in the Nordic Seas (−2–20 °C and 20–36, respectively)

were discarded. Each dive/climb cycle was subsequently inspected for density inversions, and measurements causing inversions exceeding 0.05 kg/m³ were excluded[18,32,33]. The sensors were laboratory calibrated prior to deployment. In addition, the glider data were calibrated against shipboard measurements at the times of deployment and recovery. A rendezvous partway through the deployment was used for data intercomparison. Corrections were estimated in potential conductivity–potential temperature space[34]. Offsets corresponding to 0.006 and 0.016 in salinity were applied to two of the gliders (sg559 and sg562, respectively), while a drift over the entire deployment period corresponding to 0.0008 in salinity was corrected for one glider (sg559). No corrections were applied to the conductivity sensor of the third glider (sg564) or any of the temperature sensors.

Vertical sections of potential temperature and salinity were constructed using Laplacian-spline interpolation[7,35].

Details about the August 2012 shipboard transect can be found in ref. [7] and the calibrated seal data set in ref.[21].

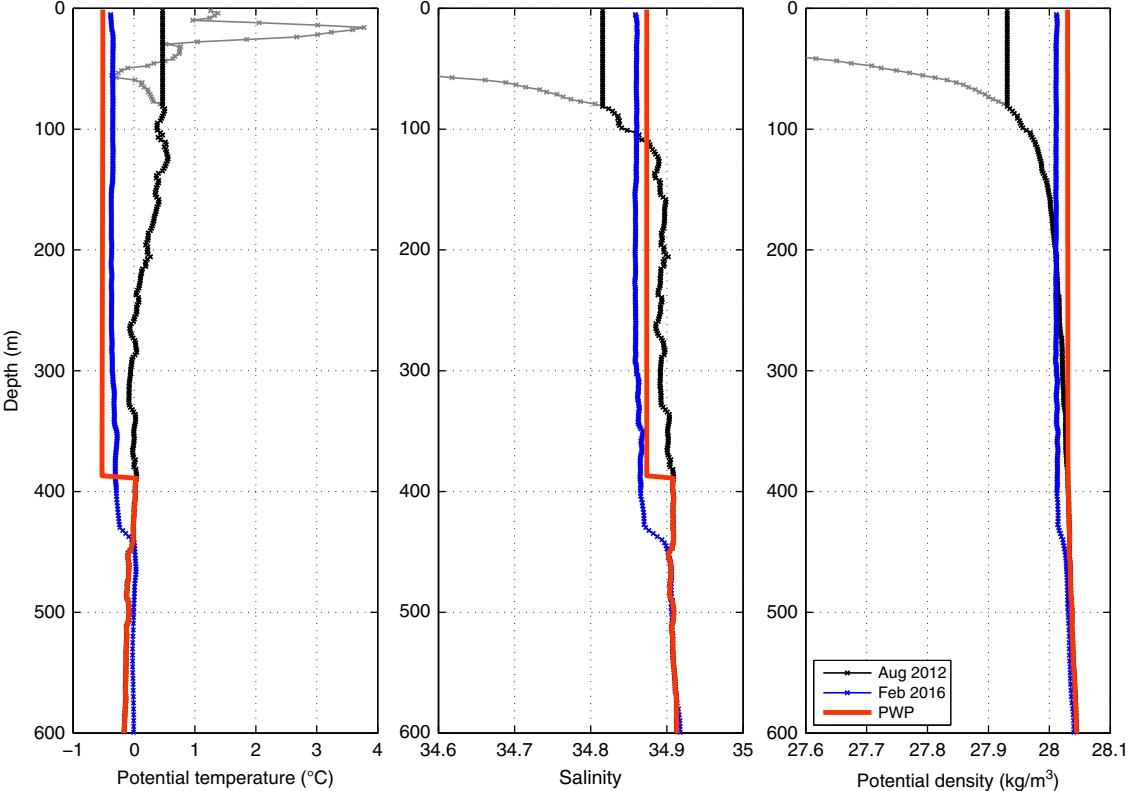

**Fig. 6** Observed and modelled hydrographic profiles. Profile from the August 2012 cruise modified to have a uniform 80 m deep mixed layer (in grey/black) that was closest to the February 2016 glider's western turnaround point (profile in blue). The red profile is the result of a simulation by the one-dimensional PWP model using the modified summertime profile as initial conditions and forced by a constant heat loss of 120 W/m² from November to mid-February

**Reanalysis data**. Wind stress, surface sensible and latent heat fluxes, as well as sea ice concentration for the climatological analyses (Fig. 5) were taken from the European Centre for Medium-Range Weather Forecasts interim reanalysis (ERA-Interim) for the period 1979–2016. Estimates of wind stress and heat fluxes were derived from short-range forecasts averaged over 6-hourly intervals (using forecast steps for 9–21 h). Only ice-free grid cells, with a sea ice concentration of less than 0.5, were considered.

**Ekman transport**. The Ekman transport distance was estimated from

$$x(\mathbf{n}) = \int dt \, \frac{1}{\rho f h} \boldsymbol{\tau}(t) \cdot \mathbf{n}_\perp, \tag{1}$$

where $\boldsymbol{\tau}(t)$ is the 6-hourly average mean wind stress in the ice-free portion of the western Iceland Sea region (Fig. 1), $\rho = 1025$ kg/m³ is the reference density of sea water, $f$ is the coriolis parameter, and $\mathbf{n}_\perp$ is the box mean unit vector perpendicular to Greenland's shelf. For the depth of the Ekman layer a constant value of $h = 50$ m was assumed, which corresponds approximately to the typical depth of the sharp pycnocline that is associated with the summertime fresh surface layer (Fig. 6). We note that the precise depth of the Ekman layer cannot be determined based on the available observations but the value of 50 m is in line with depth estimates based on the similarity height (e.g., ref. [36]). Since the Ekman transport distance is inversely proportional to Ekman depth, a decrease of $h$ by 50% would result in an increase of the estimated distance by a factor of 2, whereas an equivalent increase of $h$ would result in a lower distance by one-third.

**Mixed-layer model**. To implement the one-dimensional PWP[22] mixed-layer model, turbulent heat fluxes, which provide the dominant contribution to mixed-layer deepening[37], were imposed at the surface at each time step. Initial conditions were obtained from the August 2012 shipboard and February 2016 glider measurements. The depth and properties of the mixed layer were then adjusted until three stability criteria, chief among which is static stability, were satisfied.

**Data availability**. The glider data can be accessed at the Pangaea repository (https://doi.org/10.1594/PANGAEA.884339). The shipboard data can be obtained from http://kogur.whoi.edu. The seal data can be found in the MEOP database (www.meop.net). The ERA-Interim reanalysis data were obtained from the European Centre for Medium-Range Weather Forecasts and the passive microwave sea ice product[38] was obtained from the National Snow and Ice Data Center.

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

## Acknowledgements

We are deeply indebted to the NACO team at the University of Bergen for the successful deployment and operation of the gliders, in particular Erik Magnus Bruvik, Idar Hessevik, Karsten Kvalsund, and Tor de Lange. We thank Héðinn Valdimarsson for assistance during deployment and recovery of the gliders, and James Bennett and Bastien Queste for a very helpful tutorial on glider data processing. Dallas Murphy and Pål Erik Isachsen kindly provided valuable comments to the manuscript. We thank Kit Kovacs and Christian Lydersen, the Norwegian Polar Institute and MEOP (NFR grant number 176477), and Pål Erik Isachsen for providing access to the calibrated seal data set. Support for this work was provided by the Norwegian Research Council under Grant agreement no. 231647 (L.H. and K.V.), the Bergen Research Foundation under Grant BFS2016REK01 (K.V.), and the Centre for Climate Dynamics at the Bjerknes Centre through the FRESHWATER project (K.V.). Additional funding was provided by the Swiss National Science Foundation grants P2EZP2162267 and P300P2174307 (L.P.), the National Science Foundation grant OCE-1558742 (M.A.S.), the Norway Fulbright Foundation (K.V.), the Canada Fulbright Foundation (G.W.K.M.), and the Natural Sciences and Engineering Research Council of Canada (G.W.K.M.).

## Author contributions

K.V., L.P., and L.H. collected and analysed the data, K.V. and L.P. wrote the paper, and all authors contributed ideas, interpreted the results, and clarified the implications throughout the study.

## Additional information

**Competing interests:** The authors declare no competing interests.

