## [Peer Review File · Nature Communications]

Reviewers' comments:

Reviewer #1 (Remarks to the Author):

Vage et al. present a possible new mechanism that could impart variability on the properties of the Atlantic Meridional Overturning Circulation (AMOC) and this mechanism is a highly localized process. The AMOC is often viewed as a large-scale only flow and it has been only recently recognized that localized processes may have impacts on its variability (e.g. Lozier, 2010, DOI:10.1126/science.1189250). Furthermore, this new mechanism has a surprising implication – warmer climate and the resulting reduced sea ice coverage could result in stronger convection in the East Greenland Current. As far as I know, this mechanism is a new idea and it has the potential to influence thinking in our field.

I believe the data to be technically sound and that there is sufficient evidence supporting the two major results: 1) Atlantic-origin water in the East Greenland Current can be re-ventilated and 2) the re-ventilation rests on the confluence of ice-free conditions and strong wintertime Ekman transport. These results are shown to be broadly consistent across independent data sets. The results are also clearly presented and the manuscript is well-written.

I have some minor comments:

Lines 14-15: While I agree with the argument on lines 96-103 that the re-ventilation of Atlantic-origin water is potentially consistent with the seasonal cycle in temperature at the Denmark Straits, I believe that the statement that re-ventilation "...impacts the properties of the DSOW" must be qualified because we do not know (yet) if there is indeed a direct impact of re-ventilation on DSOW properties. To me, the wording as it is implies a direct impact, which has not been shown. The re-ventilation documented in this paper is from the winter of 2015-2016 while the seasonal cycle in Jochumsen et al. (2012) is from 1996 to 2011. For the sake of abstract-only readers, the authors should also clarify that it is the seasonal cycle (and not the interannual variability) that may be consistent with re-ventilation.

Line 37: Three gliders were deployed but only data from two gliders are presented. Also, the data presented encompass a relatively short time span (zig-zags in Fig. 2 imply 28 and 27 dives in panels b/e and c/f, respectively, so at ~6 hours/dive each = ~7 days). Given the operational capabilities of Seaglider (~6 months), are there any additional data collected in the EGC? I understand that technical or logistical (e.g. sea ice) issues could have prevented additional data collection. However, if the authors did make additional glider observations in the EGC, why are they not included here? If, for example, the data subsets presented here are the only evidence of re-ventilation and we know for certain that re-ventilation did not occur in this region otherwise, then in my opinion those other data must be discussed. EGC re-ventilation is a newly discovered mechanism and it is important to state whether it is spatially/temporally uniform or patchy, if there is data to support such a statement. Fig. 3 suggests spatial uniformity of the seasonal change in the freshwater capping layer, but that figure is limited to the upper 50 m.

Line 91: Please clarify whether the "mean November profile" is from seal data or glider data.

Lines 94-95: I see evidence that local formation is possible but I don't agree with the use of "very likely formed locally" because the authors do not consider non-local formation, so it is hard to assess likelihood.

Lines 125-128: A quantification of the deviations of the final glider data relative to the shipboard data used for calibration is needed in order to summarize the accuracy of the measurements. Given typical accuracies of Seaglider, I expect these deviations to be small relative to the signals in Fig. 2.

Figure 3: Please confirm that the horizontal error bars for January are covered up by the dot.

Reviewer #2 (Remarks to the Author):

Review by Sheldon Bacon of Våge et al. "Linking ocean convection to the recent ice edge retreat along east Greenland", submitted to Nature.

The authors' thesis, well-supported by the evidence, is that sea ice retreat allows increased air-sea heat fluxes, and that the exposure of Atlantic-origin water (with its burden of salt) to these surface fluxes enables convection (i.e. dense water formation) that stands to impact initially the waters that overflow the Greenland-Scotland Ridge, and ultimately the Atlantic Meridional Overturning Circulation itself, with consequences for climate.

I think this is an important piece of work that is well worth publishing. I have just one request to the authors. In a coincidence of timing, a related piece was recently published in Nature Climate Change by Sevellec et al., "Arctic sea ice decline weakens the AMOC" (doi:10.1038/NCLIMATE3353). This is a (climate) model study, which begins "[t]he ongoing decline of Arctic sea ice exposes the ocean to anomalous surface heat and freshwater fluxes, resulting in positive buoyancy anomalies that can affect ocean circulation". It only appeared online on 31 July of this year so please don't think I'm criticising its absence from the present manuscript. But it's out there and I think some words to "compare and contrast" are needed.

Reviewer #3 (Remarks to the Author):

Review of Vage et al., 'Linking ocean convection to the recent ice edge retreat along east Greenland'

This manuscript presents wintertime observations along the eastern coast of Greenland that convincingly document re-ventilation of Atlantic-origin waters as they transit southward toward the Denmark Strait. The primary observations, collected by autonomous underwater gliders during times when weather generally prevents ship-based sampling, provide a unique view in an under-sampled area and the one-dimensional simulations provide good insight into the dynamics at play. I have only a few relatively minor comments that should be addressed before the manuscript is accepted for publication.

1. Figure 2, which shows the key observations, is difficult to interpret. First, the color bars are highly nonlinear, which accentuates the artificial visual gradients already resulting from the jet colormap. This is likely done to highlight water masses, but results in a misleading picture of the water mass gradients. I strongly suggest use of a linear color bar and perceptually uniform colormaps. To outline particular water masses, key isotherms or isohalines can be drawn. Second, the inbound and outbound glider transects (Fig 2 b,c,e,f) are shown with their direction reversed and the inbound transects misaligned in longitude relative to the longer ship-based, summer transects (2a,d). This presentation is difficult to interpret. A two-column figure with sequential and aligned-in-longitude temperature in one column and salinity in the other would be much simpler to interpret.

2. It is apparent from close examination of Figure 2 that the glider observations have had some sort of temporal/spatial mapping/smoothing applied. There is a notable absence of profile-to-profile variability due to internal wave heave, and water properties are contoured beyond the range of sampling indicated by the schematic sawtooth (e.g., at the eastern end of panel 2c). Such mapping is fine, but the details of the mapping should be included in the methods section.

3. Line 63: The observations shown in Figure 3 do not provide support for the statements that 'a well-defined surface layer is no longer present. At that that time the water column has an ~80 m deep mixed layer and is well-preconditioned for convection.' Is there a way to indicate mixed layer depths from the seal-based and historical measurements in Fig. 3?

4. Fig. 4 caption: give years of ERA-Interim period here for clarity. The dates are given in the methods, but it would be helpful to include here parenthetically as well.

December 21, 2017

Reply to Reviewer #1 of NCOMMS-17-21056-T,
Linking ocean convection to the recent ice edge retreat along east Greenland

Vage et al. present a possible new mechanism that could impart variability on the properties of the Atlantic Meridional Overturning Circulation (AMOC) and this mechanism is a highly localized process. The AMOC is often viewed as a large-scale only flow and it has been only recently recognized that localized processes may have impacts on its variability (e.g. Lozier, 2010, DOI:10.1126/science.1189250). Furthermore, this new mechanism has a surprising implication – warmer climate and the resulting reduced sea ice coverage could result in stronger convection in the East Greenland Current. As far as I know, this mechanism is a new idea and it has the potential to influence thinking in our field.

I believe the data to be technically sound and that there is sufficient evidence supporting the two major results: 1) Atlantic-origin water in the East Greenland Current can be re-ventilated and 2) the re-ventilation rests on the confluence of ice-free conditions and strong wintertime Ekman transport. These results are shown to be broadly consistent across independent data sets. The results are also clearly presented and the manuscript is well-written.

We thank you very much for your positive and constructive review of our manuscript. Your comments are addressed below.

Lines 14-15: While I agree with the argument on lines 96-103 that the re-ventilation of Atlantic-origin water is potentially consistent with the seasonal cycle in temperature at the Denmark Straits, I believe that the statement that re-ventilation ...impacts the properties of the DSOW must be qualified because we do not know (yet) if there is indeed a direct impact of re-ventilation on DSOW properties. To me, the wording as it is implies a direct impact, which has not been shown. The re-ventilation documented in this paper is from the winter of 2015-2016 while the seasonal cycle in Jochumsen et al. (2012) is from 1996 to 2011. For the sake of abstract-only readers, the authors should also clarify that it is the seasonal cycle (and not the interannual variability) that may be consistent with re-ventilation.

This is a good point, we do not know that re-ventilation of the Atlantic-origin water in the East Greenland Current actually impacts the properties of the Denmark Strait Overflow Water. The wording has been changed to take this into account. However, in addition to modifying the seasonal cycle of the DSOW, re-ventilation could also contribute to a cooling and possible densification of the DSOW, at least for parts of the year. Hence we think that rephrasing the abstract only to take into account the lack of direct verification that re-ventilation impacts DSOW properties is most accurate.

Line 37: Three gliders were deployed but only data from two gliders are presented. Also, the data presented encompass a relatively short time span (zig-zags in Fig. 2 imply 28 and 27 dives in panels b/e and c/f, respectively, so at 6 hours/dive each = 7 days). Given the operational capabilities of Seaglider (6 months), are there any additional data collected in the EGC? I understand that technical or logistical (e.g. sea ice) issues could have prevented additional data collection. However, if the

authors did make additional glider observations in the EGC, why are they not included here? If, for example, the data subsets presented here are the only evidence of re-ventilation and we know for certain that re-ventilation did not occur in this region otherwise, then in my opinion those other data must be discussed. EGC re-ventilation is a newly discovered mechanism and it is important to state whether it is spatially/temporally uniform or patchy, if there is data to support such a statement. Fig. 3 suggests spatial uniformity of the seasonal change in the freshwater capping layer, but that figure is limited to the upper 50 m.

Unfortunately the gliders were operated primarily in the interior Iceland Sea to the east of the Kolbeinsey Ridge. The two transects presented in Fig. 2 are the only transects in which the gliders approached the ice edge and the EGC. We have now specified this in the text and included the complete glider trajectories in Figure 1. Hence the glider data cannot inform us about the variability of re-ventilation. The seal data set, which has ok resolution in the upper 100-200 m and few measurements at depth, is also not well-suited for that purpose.

Line 91: Please clarify whether the “mean November profile” is from seal data or glider data.

We state in the text that the mean November profile was obtained from the combined seal and historical hydrographic data set.

Lines 94-95: I see evidence that local formation is possible but I don't agree with the use of “very likely formed locally” because the authors do not consider non-local formation, so it is hard to assess likelihood.

This is a valid point. We have modified the text to state that the deep mixed layers recorded by the glider are consistent with local formation.

Lines 125-128: A quantification of the deviations of the final glider data relative to the shipboard data used for calibration is needed in order to summarize the accuracy of the measurements. Given typical accuracies of Seaglider, I expect these deviations to be small relative to the signals in Fig. 2.

The details of the corrections applied to the glider data have been included in the Methods section. The deviations are indeed small relative to the signals in Fig. 2

Figure 3: Please confirm that the horizontal error bars for January are covered up by the dot.

The horizontal error bars are covered by the dot. Only three profiles were available for that month. By chance, all of these had very similar upper-ocean salinities, so the standard error of the measurements became very small.

December 21, 2017

Reply to Reviewer #2 of NCOMMS-17-21056-T,
Linking ocean convection to the recent ice edge retreat along east Greenland

Review by Sheldon Bacon of Våge et al. “Linking ocean convection to the recent ice edge retreat along east Greenland”, submitted to Nature.

The authors’ thesis, well-supported by the evidence, is that sea ice retreat allows increased air-sea heat fluxes, and that the exposure of Atlantic-origin water (with its burden of salt) to these surface fluxes enables convection (i.e. dense water formation) that stands to impact initially the waters that overflow the Greenland-Scotland Ridge, and ultimately the Atlantic Meridional Overturning Circulation itself, with consequences for climate.

I think this is an important piece of work that is well worth publishing. I have just one request to the authors. In a coincidence of timing, a related piece was recently published in Nature Climate Change by Sevellec et al., “Arctic sea ice decline weakens the AMOC” (doi:10.1038/NCLIMATE3353). This is a (climate) model study, which begins “[t]he ongoing decline of Arctic sea ice exposes the ocean to anomalous surface heat and freshwater fluxes, resulting in positive buoyancy anomalies that can affect ocean circulation”. It only appeared online on 31 July of this year so please don’t think I’m criticising its absence from the present manuscript. But it’s out there and I think some words to “compare and contrast” are needed.

Dear Dr. Bacon.

Thank you very much for the positive review of our manuscript and for pointing out the recent publication by Sévellec et al. In the final paragraph we have added a statement to the effect that as large-scale sea ice loss may lead to a decline of the AMOC (Sévellec et al., 2017), the increased ventilation of Atlantic water as a result of the ice edge retreat toward Greenland is unexpected.

December 21, 2017

Reply to Reviewer #3 of NCOMMS-17-21056-T,
Linking ocean convection to the recent ice edge retreat along east Greenland

Review of Vage et al., “Linking ocean convection to the recent ice edge retreat along east Greenland”

This manuscript presents wintertime observations along the eastern coast of Greenland that convincingly document re-ventilation of Atlantic-origin waters as they transit southward toward the Denmark Strait. The primary observations, collected by autonomous underwater gliders during times when weather general prevents ship-based sampling, provide a unique view in an under-sampled area and the one-dimensional simulations provide good insight into the dynamics at play. I have only a few relatively minor comments that should be addressed before the manuscript is accepted for publication.

We thank you very much for your positive and constructive review of our manuscript. Your comments are addressed below.

1. Figure 2, which shows the key observations, is difficult to interpret. First, the color bars are highly nonlinear, which accentuates the artificial visual gradients already resulting from the jet colormap. This is likely done to highlight water masses, but results in a misleading picture of the the water mass gradients. I strongly suggest use of a linear color bar and perceptually uniform colormaps. To outline particular water masses, key isotherms or isohalines can be drawn. Second, the inbound and outbound glider transects (Fig 2 b,c,e,f) are shown with their direction reversed and the inbound transects misaligned in longitude relative to the longer ship-based, summer transects (2a,d). This presentation is difficult to interpret. A two-column figure with sequential and aligned-in-longitude temperature in one column and salinity in the other would be much simpler to interpret.

It is true that the non-linear color bars accentuate visual gradients. However, in particular for salinity, the range of observed values is very high, while the water masses of primary interest at intermediate depth have very slight differences in salinity. For that reason linear color maps do not work well in this region, hence most of the recent studies from this region have utilized non-linear color bars (e.g. Våge et al., 2013; Harden et al., 2016; Håvik et al., 2017; Mastropole et al., 2017; Pickart et al., 2017). We think that the non-linear color bar without perceptually uniform colormap is best suited for the presentation of our results.

We understand that the figure may be difficult to interpret. In order to aid the reader, we have modified the figure such that the lower horizontal axis now shows time, which makes it more clear how the glider moves. The upper horizontal axis shows longitude, which helps relate the glider measurements to the shipboard transect. We prefer this layout to the suggested two-column figure, which would have increased the number of panels to 10 instead of the 6 currently contained in Figure 2.

2. It is apparent from close examination of Figure 2 that the glider observations have had some sort of temporal/spatial mapping/smoothing applied. There is a notable absence of profile-to-profile variability due to internal wave heave, and water properties are contoured beyond the range of

sampling indicated by the schematic sawtooth (e.g., at the eastern end of panel 2c). Such mapping is fine, but the details of the mapping should be included in the methods section.

The details and references of the mapping procedure have been included in the Methods section.

3. Line 63: The observations shown in Figure 3 do not provide support for the statements that “a well-defined surface layer is no longer present. At that that time the water column has an 80 m deep mixed layer and is well-preconditioned for convection.” Is there a way to indicate mixed layer depths from the seal-based and historical measurements in Fig. 3?

Due primarily to low vertical resolution, the seal-based profiles, which constitute the bulk of the wintertime measurements, are not well suited for determination of mixed-layer depth. The addition of mixed-layer depths to the figure would make it rather busy and more difficult to interpret. As such, we think that the best use of the combined seal and historical data set is to shed light on the seasonal cycle of the upper layer.

4. Fig. 4 caption: give years of ERA-Interim period here for clarity. The dates are given in the methods, but it would be helpful to include here parenthetically as well.

The years of the ERA-Interim period have been added to the caption.

References

- Harden, B. E., Pickart, R. S., Valdimarsson, H., Richards, C., Våge, K., de Steur, L., Bahr, F., Torres, D. J., Børve, E., Jónsson, S., Macrander, A., Østerhus, S., Håvik, L., Hattermann, T., 2016. Upstream sources of the Denmark Strait Overflow: observations from a high-resolution mooring array. *Deep Sea Research I* 112, 94–112, doi:10.1016/j.dsr.2016.02.007.
- Håvik, L., Pickart, R. S., Våge, K., Thurnherr, A. M., Beszczynska-Möller, A., Walczowski, W., von Appen, W. J., 2017. Evolution of the East Greenland Current from Fram Strait to Denmark Strait: Synoptic measurements from summer 2012. *Journal of Geophysical Research: Oceans*, doi:10.1002/2016JC012228.
- Mastropole, D., Pickart, R. S., Valdimarsson, H., Våge, K., Jochumsen, K., Girton, J., 2017. On the hydrography of Denmark Strait. *Journal of Geophysical Research: Oceans* 122, 306–321, doi:10.1002/2016JC012007.
- Pickart, R. S., Moore, G. W. K., Torres, D. J., Våge, K., Valdimarsson, H., Nobre, C., Moore, G. W. K., Jónsson, S., Mastropole, D., 2017. The North Icelandic Jet and its relationship to the North Icelandic Irminger Current. *Journal of Marine Research* 75, 605–639, doi:10.1357/002224017822109505.
- Våge, K., Pickart, R. S., Spall, M. A., Moore, G. W. K., Valdimarsson, H., Torres, D. J., Erofeeva, S. Y., Nilsen, J. E. Ø., 2013. Revised circulation scheme north of the Denmark Strait. *Deep Sea Research I* 79, 20–39, doi:10.1016/j.dsr.2013.05.007.

REVIEWERS' COMMENTS:

Reviewer #1 (Remarks to the Author):

The authors have addressed my comments and I have nothing to add relative to my previous review. It's a solid paper that I enjoyed reading. I believe it makes a worthy contribution to our field.

Reviewer #3 (Remarks to the Author):

Rereview of Vage et al., 'Linking ocean convection to the recent ice edge retreat along east Greenland'

Though I still find Figure 2 challenging to interpret and stand by my previous suggestions about presentation in that figure, I believe the authors have addressed the substantial points raised in my previous review as well as the other reviewers. So, I recommend publication with any further stylistic changes left to the editor and authors to settle on.